# Effects of Organic Acids on the Release of Fruity Esters in Water: An Insight at the Molecular Level

**DOI:** 10.3390/molecules27092942

**Published:** 2022-05-04

**Authors:** Yu Liu, Hui Xi, Yingjie Fu, Peng Li, Shihao Sun, Yongli Zong

**Affiliations:** 1Key Laboratory in Flavor & Fragrance Basic Research, Zhengzhou Tobacco Research Institute, China National Tobacco Corporation, Zhengzhou 450001, China; liuyuing@sust.edu.cn (Y.L.); xihui1316@126.com (H.X.); fyjztri@163.com (Y.F.); lipeng@ztri.com.cn (P.L.); 2Center of Advanced Analysis and Gene Sequencing, Zhengzhou University, Zhengzhou 450001, China

**Keywords:** esters, organic acids, aroma release, density functional theory, intermolecular interaction, odour detection threshold

## Abstract

It is well known that organic acids (OAs) could affect the flavour of fruit juices and beverages. However, the molecular mechanism of aroma release is still unclear. In this study, the effects of citric acid (CA), L-(-)-malic acid (MA) and L-lactic acid (LA) on the release of six selected esters and their sensory perception were investigated by means of HS-GC-MS analyses and odour detection threshold determination, respectively. Meanwhile, the density functional theory (DFT) calculation was employed to explore the interaction modes between esters and OAs. HS-GC-MS analyses showed that the concentration and the type of OAs regulated the release of esters. The results were basically consistent with the detection threshold change of those esters. The DFT calculation suggested that the main intermolecular interaction was hydrogen bonds, and several esters could form a ternary ring structure with OAs through hydrogen bonds. The interactions can induce the different release behaviours of esters in OAs water solution. The number of carboxyl functional groups in OAs and the spatial conformation of esters appeared to influence the magnitude of the interaction. The above results demonstrated the mechanism of OAs affecting the release of esters and indicated a possible flavour control way by using different OAs and OA concentrations.

## 1. Introduction

Aroma is one of the most essential characteristics of food and beverage, determining consumer acceptability. Despite many aroma compounds having been identified and studied, a thorough understanding of aroma compound release is required to obtain a high quality of food aroma products. Aroma release is highly sensitive to numerous factors [1,2,3,4], and the food or beverage composition play a vital role [5,6]. Non-volatile matrix components, such as sugars, polyphenols and OAs, have been found to influence aroma release [7,8,9,10,11,12,13]. However, the understanding of the interaction mechanism is unclear. Both the physiochemical properties of aroma compounds and the interaction or binding between the aroma component and the non-volatile matrix components significantly influence aroma release and perception [14,15]. It is meaningful to comprehend the interaction or releasing mechanism of flavour compounds in food and beverage due to the possible flavour modulation and improvement in sensory properties.

The perception of volatiles in the sensory analysis is closely related to the aroma distribution between the gas and liquid phase, which may be related to the perception during food consumption [16,17,18]. Determined by the vapor-liquid equilibrium of each odorant, the distribution is characterised by the partition coefficient (*k*). Because of the lack of calibration curve, simple sample preparation and good reproducibility, the Phase Ratio Variation (PRV) [19,20] method has become a popular method to determine the partition coefficient of aroma compounds [20,21,22,23,24,25,26].

This study aimed to investigate the role of OAs in ester solution and explore the mechanism insight into molecular level. Firstly, the release of fruity esters was studied with/without individual CA, MA and LA at three different concentrations, which was measured by calculating their partition coefficient and retention value (*R* value). Furthermore, the odour detection threshold was evaluated to explain the odour regulation effect of OAs. Meanwhile, theoretical calculation based on DFT was employed to determine the interaction modes between OAs and esters and clarify the mechanism of the changed aroma release.

## 2. Results and Discussion

### 2.1. Interactions between Esters and OAs

Since the interactions between esters and OAs are critical to the release of esters, theoretical calculation was carried out to investigate their possible interaction modes. The weak interactions between molecules could be visually seen from the IGM color map (Figure 1, Figure 2 and Figure 3). The type and relative intensity of interactions could be judged according to the color of the diagram. The dark blue region has strong attraction, which are generally hydrogen bonds, strong halogen bonds and so on; the green region shows weak attraction, which is van der Waals force; the red region shows strong mutual exclusion, which is generally the steric hindrance effect in the ring or cage. There were van der Waals force and hydrogen bonds between esters and OAs. The hydroxyl or (and) carbonyl groups of OAs participated in hydrogen bond formation. It is interesting to note that several esters formed a ternary ring with OAs molecules by hydrogen bonds, such as butyl acetate and CA, ethyl valerate and MA, ethyl isobutyrate and LA. In addition, steric hindrance was also observed for ethyl isovalerate and CA. As for MA and LA, there is no direct interaction (hydrogen bond or van der Waals force) between certain esters and one of the MA or LA, such as ethyl butyrate and MA, ethyl isovalerate and LA. Meanwhile, there is a higher probability of steric hindrance between branched esters and OAs. Although CA, LA and MA have similar chemical structure, the OAs-esters interaction differs significantly, which may be related to the number of carboxyl functional groups of OAs and the spatial configuration of esters.

### 2.2. Release of Esters in Aqueous Solutions

Partition coefficients were calculated for the selected esters (propyl acetate, butyl acetate, ethyl butyrate, ethyl isobutyrate, ethyl valerate and ethyl isovalerate) in aqueous solution alone or supplemented with CA, MA and LA at three different concentrations, respectively (Appendix A). Generally, the partition coefficients of esters in water highly depend on their hydrophobicity and vapor pressure [27]. The more hydrophobic the compounds, the higher the *k* value they had. The least hydrophobic compound (propyl acetate with log *P* = 1.26) was retained to a larger extent because of its greater affinity for water. It is worth noting that butyl acetate had a bigger *k* value than ethyl isobutyrate and ethyl butyrate in water. Table 1 showed that ethyl isobutyrate and ethyl butyrate possessed higher vapor pressure in water than butyl acetate, which could explain the behaviour of these compounds. This result was consistent with the previous study [27].

### 2.3. Release of Esters in OAs Solution

#### 2.3.1. Effect of Citric Acid Concentration

The influence of OAs on the release of esters was assessed in 1 g/L, 2.5 g/L and 5 g/L of OAs solution (low concentration, medium concentration and high concentration), and *R* value was utilised to indicate the percentage of retention. In 1 g/L of CA solution, all the studied esters were better retained (3.88-28.5%) than in water except ethyl isovalerate (−2.30%), showing that interactions between CA and aroma compounds have occurred (Figure 4). When CA is dissolved in water, part of CA is the dissociated form [28]. The dissociated form is more reactive, with esters inducing the reduced release of esters, and the pH hardly affect the release of esters [7,29]. Meanwhile, the two hydrogen bonds between ethyl isovalerate and CA also contributed to the behaviour. Ethyl isovalerate is a branched ester with larger molecular weight, and there was obvious steric hindrance holding back intermolecular interactions to a certain degree. Thus, the extra CA could bind free water, inducing a minor effect on salting out of ethyl isovalerate [30]. As for propyl acetate, butyl acetate and ethyl isobutyrate, there were also two hydrogen bonds between certain ester and CA (Appendix A), but they were better retained than ethyl isovalerate. The ternary ring formed by butyl acetate and ethyl isobutyrate, respectively, may be responsible for the phenomenon. Two CA molecules did not interact with propyl acetate directly in the cluster, indicating that free CA could interact with more propyl acetate. More hydrogen bonds caused the higher *R* value of ethyl valerate and ethyl butyrate than ethyl isovalerate. It is notable that the logarithmic value of *R* values of the five retained esters showed a linear relationship with log *P*. Although the regression coefficient is not significant in a strict sense, there is a trend showing that the *R* value correlates with the log *P* (Figure 5). The higher the log *P* value of ester compounds, the higher *R* values they had.

When increasing CA concentration, the release of propyl acetate, ethyl butyrate, ethyl isobutyrate, ethyl valerate and ethyl isovalerate increased significantly between low and high concentrations of CA solutions (*p* < 0.05), but the release in medium concentration CA solution was not significantly different from both low and high concentrations of CA solution. In addition, the release of butyl acetate only showed an increasing trend, and there was no significant difference. The increased release of esters may be dominated by the weak surface activity of CA and CA-water interaction. Previous studies have found that adding a low surfactant to the solution could enhance mass transfer and alter the release of aroma compounds [31,32,33]. Meanwhile, increasing CA concentration could induce more dissociated CA and CA-water interaction [30]. In the concentration range we used, hydration number increased with increasing CA concentration, resulting in the salting-out effect. Although more dissociated CA retained esters, it is not enough to counteract the first two effects. A. Hansson et al. also observed the same trend of isopentyl acetate and ethyl hexanoate [6]. However, Zhang et al. found increasing CA concentration regulated the release of ethyl acetate, ethyl butanoate, methyl hexanoate, but in the presence of pectin [29]. Moreover, the release of ethyl valerate and ethyl isovalerate were the most sensitive to increased CA concentration, which may be associated with the ternary ring structure formed by esters and CA. Ethyl valerate did not form the ternary ring. One of the hydrogen bonds of the ternary ring structure formed by ethyl isovalerate and CA was relatively weak, and there was obvious steric hindrance. Although propyl acetate did not form the ternary ring structure, two CA did not interact with hydrophobic propyl acetate. The “free” CA may aggregate and form clusters, which may generate release retention [30]. Previous studies have shown that self-aggregated tannin resulted in aroma release retention [34].

#### 2.3.2. Effect of Lactic Acid Concentration

The release of each compound was retained in 1 g/L of LA solution except ethyl butyrate (Figure 6). Compared to other esters, ethyl butyrate (−6.90%) was not bound by hydrogen bonds (Appendix A), which may be a dominating factor for the increasing release of ethyl butyrate. The free LA molecule and one hydrogen bond caused better retention of propyl acetate than ethyl butyrate, ethyl isovalerate and ethyl valerate. Ethyl isobutyrate was the only one that formed the ternary ring structure with LA, which may promote ethyl isobutyrate to be the most retained ester (22.3%).

With increasing LA concentration, ester release showed a downward trend except for propyl acetate and ethyl butyrate, which was quite different from CA. There was no significant change in release of esters except for ethyl isovalerate and ethyl butyrate. The low molecular weight of LA might be responsible for the retained esters release. In the same concentration of OAs solution, the number of LA molecules is about twice that of CA. Accordingly, there were more dissociated LA retaining esters. According to a previous report, the LA monomer was not the dominating species in solution, but the LA dimer AA-(H_2_O)_6_ [35]. Although the LA dimer could bind free water, the retention effect induced by dissociated LA and the self-aggregation dominated.

#### 2.3.3. Effect of Malic Acid Concentration

The release of each compound was retained in 1 g/L of MA solution (Figure 7), while the release of butyrate acetate was almost unchanged (−0.332%). Compared to other esters, there was only one hydrogen bond between butyrate acetate and MA (Appendix A). More retention of butyl acetate and propyl acetate were due to more hydrogen bonds. Ethyl isobutyrate (15.2%) and ethyl valerate (11.7%) were the two most retained esters, reiterating the importance of the ternary ring structure for esters’ retention.

Increasing the MA concentration increased the release of esters. The reason for this phenomenon may be the surface activity of MA and MA-water interaction.

### 2.4. Odour Detection Thresholds of Esters and Esters Added with Organic Acids

The odour detection threshold could indirectly suggest CA’s impact on aroma release and aroma-OAs interaction. To get deeper insight into the volatile-OA interaction, ester odour detection thresholds were determined in the absence or presence of OAs (Table 1). The low concentration (1 g/L) of OAs resulted in higher odour detection thresholds for the majority of the odorants. The most variable odour detection thresholds of odorants were found in 5 g/L of MA solution. The threshold value increased by 3.64-fold for ethyl isovalerate, 1.54-fold for ethyl valerate, 1.06-fold for propyl acetate and decreased by 2.27 for ethyl isobutyrate, 1.81-fold for butyl acetate and 1.04-fold for ethyl butyrate.

The threshold changes of the majority of the esters were consistent with the HS-GC-MS analyses. When there was a higher partition coefficient of ester, the odour detection trends to decrease. For instance, the *R* value of ethyl valerate and butyl acetate in 1 g/L of CA solution were 28.4% and 7.92%, respectively. The ethyl valerate odour detection threshold increased by 3.49-fold and butyl acetate odour detection threshold increased by 2.55-fold in 1 g/L of CA solution. This trend reflects the aroma retention phenomenon in CA solution through CA-aroma interaction and consists of HS-GC-MS analyses. A previous study determined the importance of the matrix choice in the determination of odour detection threshold, which could lead to over- or underestimation of threshold [36]. In addition, ethyl isobutyrate as well as ethyl isovalerate thresholds were scarcely influenced in 1 g/L of CA solution since their increasing threshold factor was lower than the dilution step (2-fold). However, Lorrain et al. found ethyl isobutyrate thresholds were not affected by the presence of catechin, but triangle tests were positive [37]. They thought it was because the value (5–9 μg/L) of thresholds were too low, but Pineau et al. and Villamor et al. found *β*-damascenone odour thresholds (50–7000 ng/L) and eugenol odour thresholds (0.89–8900 ng/L) varied over a wide range in different matrices, respectively [34,38]. Although the odour detection threshold could not show the magnitude of the interaction between volatiles and OAs, it serves as lateral evidence to reflect the intermolecular interaction in the system.

## 3. Materials and Methods

### 3.1. Chemicals

Standard-grade purity compounds were obtained from commercial sources as follows: propyl acetate and ethyl valerate from Shanghai Aladdin Biochemical Technology CO., Ltd. (Shanghai, China); butyl acetate, ethyl 2-methylpropanoate and ethyl isovalerate from TCI Development Co., Ltd. (Shanghai, China); ethyl butyrate, citric acid, L-(-)-malic acid and L-lactic acid from Sigma-Aldrich (St. Louis, MO, USA). The purity of all compounds is above 99%. The concentrations of esters are shown in Table 2.

### 3.2. Experimental Methods

#### 3.2.1. Chromatographic Conditions

The partition coefficients were determined using static headspace coupled to GC-MS. A capillary DB-5MS column was employed (length: 60 m, internal diameter: 0.25 mm, film thickness: 0.25 μm). Samples were added into glass vials (22.8 mL) under equilibrium conditions at 29 ± 1 °C, and each sample was placed stably to reach thermodynamic equilibrium. A 750-μL sample of headspace was withdrawn using a 2.5-mL thermostatic gastight syringe preheated to 35 °C on a Gerstel autosampling device, and each vial was analysed once. Gas chromatography analyses were carried out on an 8890 coupled to a 5977B quadrupole mass spectrometer (Agilent). Injections were in splitless mode, using a 4 mm i.d. deactivated gooseneck splitless liner transfer. The oven temperature was programmed from 50 °C (for 1 min) to 250 °C (for 1 min) at 20 °C/min. The carrier gas was Helium with a constant flow of 1 mL/min for the column. The mass spectrometer was operated in electron ionisation mode at 70 eV in selected-ion-monitoring mode.

#### 3.2.2. Calculation of Partition Coefficients

The partition coefficient describes the ratio of concentrations between the gas phase (*C*_gas_) and the liquid matrix (*C*_liq_) of volatile compounds at the thermodynamic equilibrium:(1)$$k=\frac{C_{\mathrm{gas}}}{C_{\mathrm{liq}}}$$

According to the PRV method developed by Ettre et al. [21], the concentration of volatiles in the headspace is proportional to the sample volume in the vial, by which the partition coefficient was determined.
(2)$$\frac{1}{A}=\frac{1}{f_{i}{\;\times \;C}_{i}^{\mathrm{liq}}\;\times \;k}+\frac{1}{f_{i}{\;\times \;C}_{i}^{\mathrm{liq}}}\;\times \;\beta$$

*A* is the chromatographic peak area at the thermodynamic equilibrium, *f_i_* is the detector response factor, $C_{i}^{\mathrm{liq}}$ is the initial concentration of the compound in the vial and *β* is the ratio between headspace (*V*_g_) and liquid (*V*_l_) volumes. Variables in this equation are *β* and *A*, and thus we can get a linear relationship by plotting 1/*A* against *β*, as follows:(3)$$\frac{1}{A}=a+b\beta$$
where $a=\frac{1}{f_{i}\;\times \;C_{i}^{\mathrm{liq}}\;\times \;k}$ and $b=\frac{1}{f_{i}\;\times \;C_{i}^{\mathrm{liq}}}$.

The partition coefficient value is obtained by plotting 1/*A* against β in the linear zone. In this work, six different volumes of solutions (0.05, 0.1, 0.5, 1, 1.5 and 2 mL) were added into glass vials with phase ratios from 446 to 10.4.

Moreover, the percentage of R value can be calculated:(4)$$R(\%)=\left(1-\frac{k_{2}}{k_{1}}\right)\;\times \;100$$

#### 3.2.3. Determination of Odour Detection Thresholds

The odour detection threshold is the lowest intensity of a sensory stimulus that has a probability of detection of 0.5. The odour detection thresholds of aroma compounds were determined in water with or without individual OAs at 1, 2.5 and 5 g/L (low, medium, and high concentration) with an ascending procedure and the three-alternative forced choice presentation method (3-AFC). The best estimate threshold was calculated as the geometric mean of the highest concentration missed, and the next higher concentration [42]. To guarantee data obtained under the same condition, odour detection thresholds of aroma alone and aroma added with OAs were determined in 3 days (10 sessions). Panelists (5 male and 5 females, aged from 23 to 36) were selected for their experience.

#### 3.2.4. Theoretical Methods

The workflow of theoretical calculation is shown in Figure 8. A conformational search was performed to determine the primary conformation by Molclus [43] and Gaussian 16 software [44]. The M06-2X functional was adopted for all calculations [45]. The def2-SV(P) basis set was used for geometry optimisation and frequency calculation [46], and the optimal geometry for each compound was determined. The DFT-D3 dispersion correction was applied to correct the weak interaction to improve the calculation accuracy [47]. The nature of nonvalent interaction was studied by means of the Independent Gradient Model (IGM) method [48] through Multiwfn software [49]. The IGM diagram is rendered by VMD [50]. To get close to the reality of the experiment, the theoretical calculation is based on the ratio of ester to organic acid at 1:5.

### 3.3. Statistical Analysis

The data were analysed using the Kruskal-Wallis nonparametric test (SPSS, version 23, Chicago, IL, USA). The statistically significant level was 5% (*p* < 0.05).

## 4. Conclusions

This work systemically studied the mechanism of influence of OAs on the release of esters and detection thresholds, which was mainly due to the intermolecular interaction. OAs had variable effects on the release of esters and their detection thresholds, implying the type and concentration of OAs were important factors of flavour release. DFT calculation indicated that the intermolecular interaction between OAs and esters were mainly hydrogen bonds. The hydrogen bonds and the ternary ring structure presented in OAs-esters systems induced the different release behaviours of esters in OAs water solution. The spatial conformation of esters and the number of carboxyl groups of OAs could impact the magnitude of the interaction. Moreover, the self-aggregation and surface activity of OAs may also regulate the release by impacting the intermolecular interaction. The disclosed mechanism of ester release has significant instruction for flavour control and production practice.

## Figures and Tables

**Figure 1 molecules-27-02942-f001:**
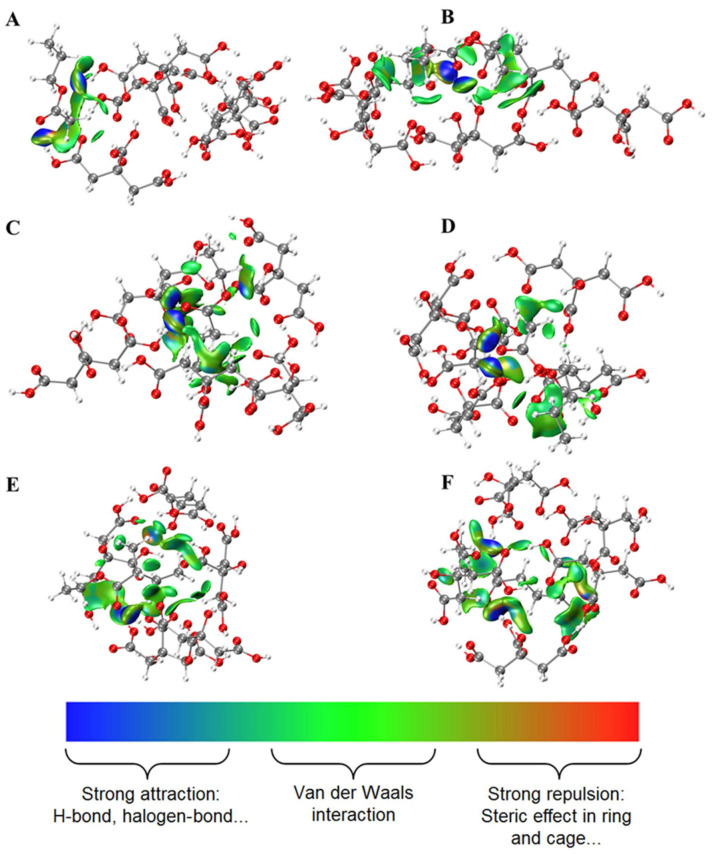
Weak interactions between esters and citric acid. (**A**) C2C3, (**B**) 2MeC3C2, (**C**) C4C2, (**D**) C2C4, (**E**) 3MeC4C2, (**F**) C5C2, H-white, C-gray, O-red.

**Figure 2 molecules-27-02942-f002:**
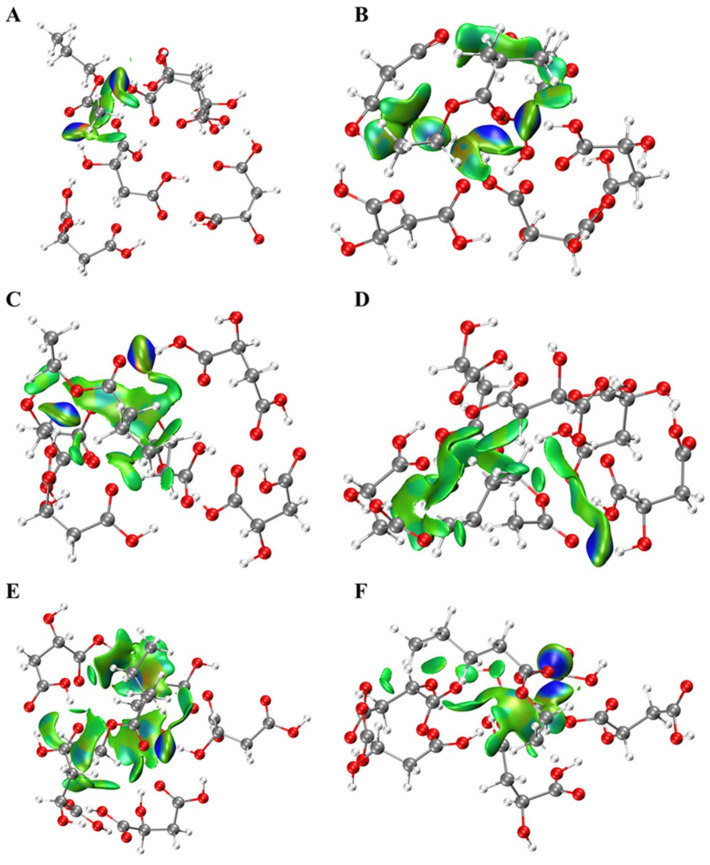
Weak interactions between esters and malic acid. (**A**) C2C3, (**B**) 2MeC3C2, (**C**) C4C2, (**D**) C2C4, (**E**) 3MeC4C2, (**F**) C5C2, H-white, C-gray, O-red.

**Figure 3 molecules-27-02942-f003:**
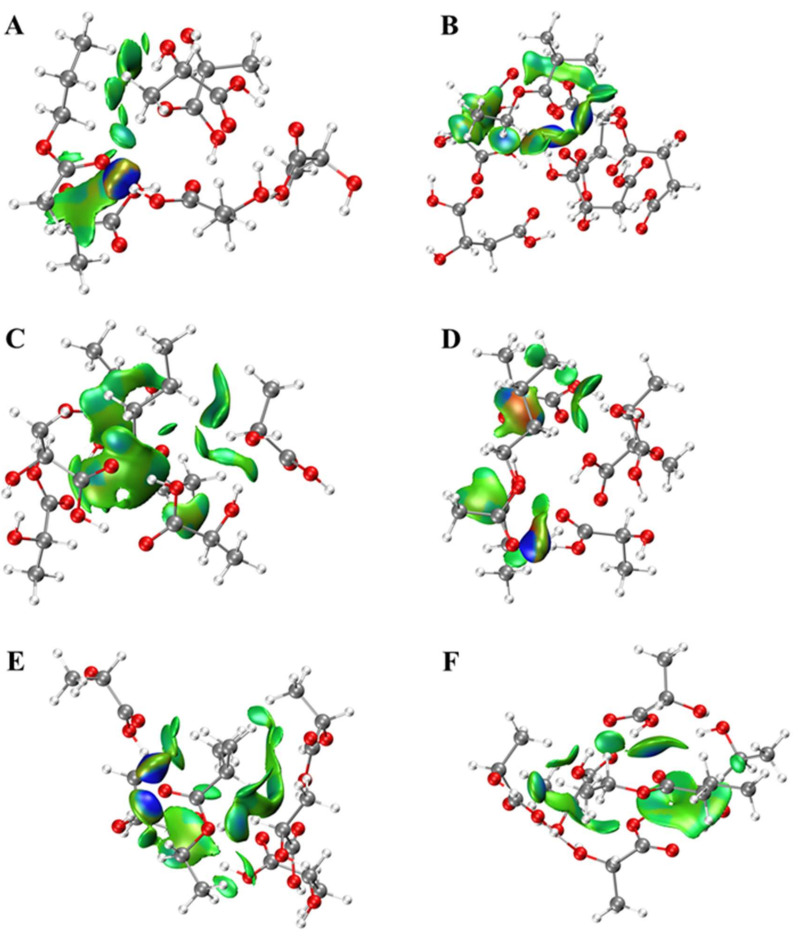
Weak interactions between esters and lactic acid. (**A**) C2C3, (**B**) 2MeC3C2, (**C**) C4C2, (**D**) C2C4, (**E**) 3MeC4C2, (**F**) C5C2, H-white, C-gray, O-red.

**Figure 4 molecules-27-02942-f004:**
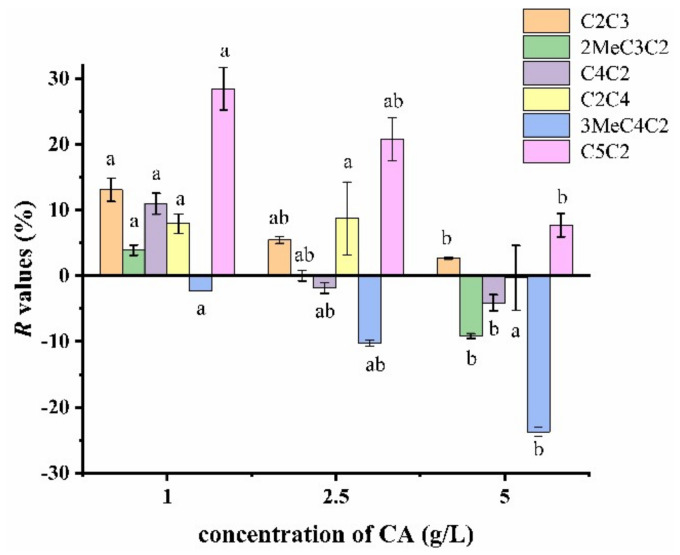
*R* values of esters in 1, 2.5 and 5 g/L of CA solution. Different letters indicate significant difference for each ester (*p* < 0.05).

**Figure 5 molecules-27-02942-f005:**
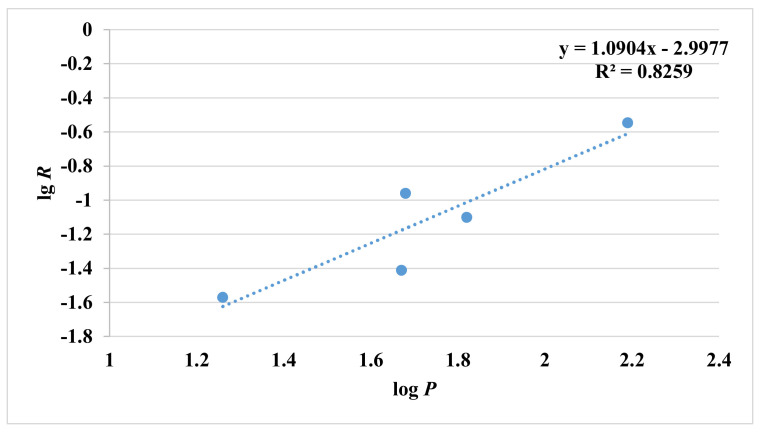
Relationship between logarithmic value of *R* values and log *P* of five retained esters in 1 g/L of CA solution.

**Figure 6 molecules-27-02942-f006:**
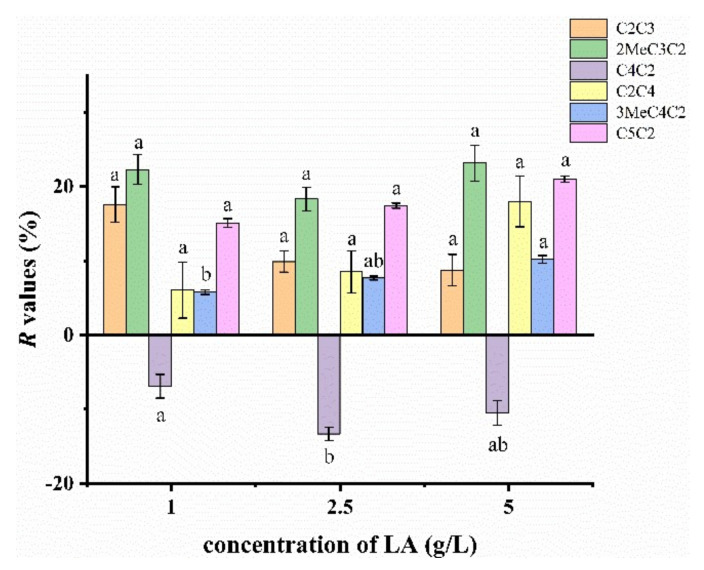
*R* values of esters in 1, 2.5 and 5 g/L of LA solution. Different letters indicate significant difference for each ester (*p* < 0.05).

**Figure 7 molecules-27-02942-f007:**
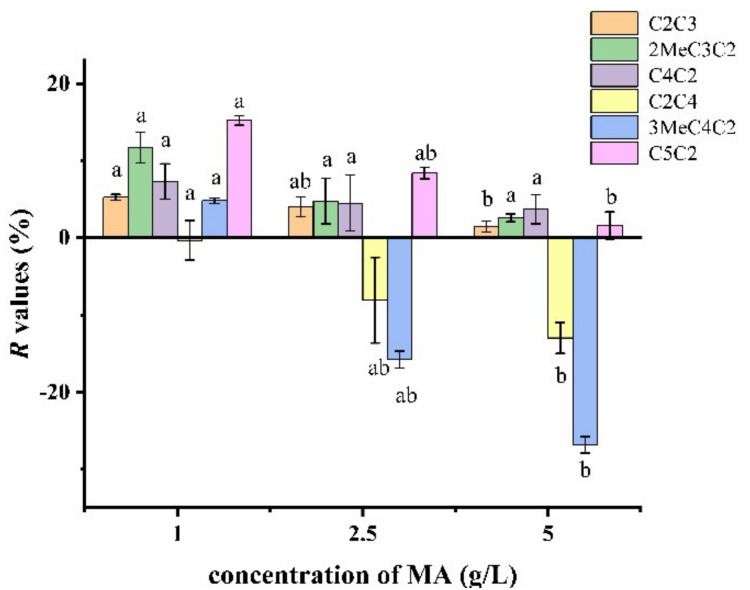
*R* values of esters in 1, 2.5 and 5 g/L of MA solution. Different letters indicate significant difference for each ester (*p* < 0.05).

**Figure 8 molecules-27-02942-f008:**
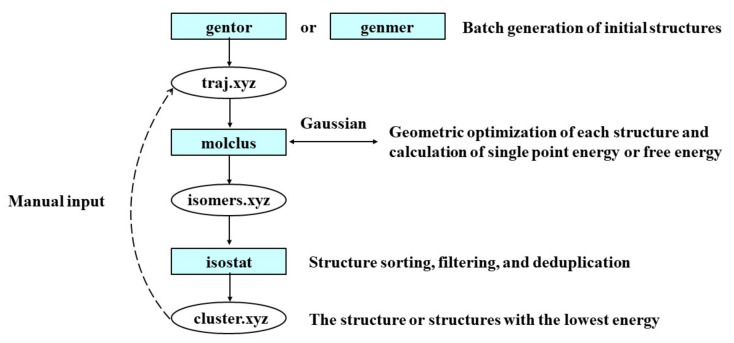
The workflow of theoretical calculation.

**Table 1 molecules-27-02942-t001:** Odour detection threshold of esters and esters added with organic acid.

	Odour Detection Threshold (μg/L) in Solution									
	Water	1 g/L CA	2.5 g/L CA	5 g/L CA	1 g/L LA	2.5 g/L LA	5 g/L LA	1 g/L MA	2.5 g/L MA	5 g/L MA
Propyl acetate	2.27	4.19	2.74	2.49	6.12	3.98	3.46	2.74	2.84	2.41
Butyl acetate	68.0	80.4	93.1	57.0	136	111	136	64.0	33.9	37.5
Ethyl butyrate	0.691	1.76	0.614	0.42	0.442	0.345	0.425	0.817	0.781	0.666
Ethyl isobutyrate	0.262	0.249	0.249	0.131	0.621	0.552	0.584	0.552	0.395	0.276
Ethyl valerate	18.2	63.5	55.0	34.4	46.7	58.9	47.6	46.7	34.4	28.1
Ethyl isovalerate	0.201	0.184	0.0884	0.0582	0.276	0.260	0.295	0.276	0.0650	0.0552

**Table 2 molecules-27-02942-t002:** Physicochemical properties of esters.

Compound	Aromatic Descriptor ^1^	Abbreviation	Water Solubility ^2^ (g/L)	Log *P* ^3^	Vapor Pressure ^2^ (mm Hg)	Concentration (μg/L)
Propyl acetate [39,40]	Celery, floral, pear	C2C3	10.670	+1.26	35.1	500
Ethyl isobutyrate [39,41]	Kiwi, strawberry, solvent	2MeC3C2	3.172	+1.67	24.2	400
Ethyl butyrate [39,40]	Apple, butter, cheese	C4C2	2.745	+1.68	14.6	400
Butyl acetate [39]	Apple, banana	C2C4	3.128	+1.82	11.9	400
Ethyl isovalerate [41]	Apple, fruit, pineapple	3MeC4C2	1.070	+1.9	7.98	500
Ethyl valerate [39]	Apple, dry fish, herb	C5C2	0.926	+2.19	4.8	500

^1^ Aromatic descriptor from the Flavor Ingredient Library (www.femaflavor.com, accessed on 20 April 2022). ^2^ Vapor pressure (at 25 °C) and water solubility (at 25 °C): from EPI suit 4.1 calculation. ^3^ Log *P* obtained from online Molinspiration cheminformatics services (www.molinspiration.com, accessed on 3 January 2022).

## Data Availability

The data that support the findings of this study are available from the corresponding author upon reasonable request.

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
