# Peer review of "Effects of Organic Acids on the Release of Fruity Esters in Water: An Insight at the Molecular Level"

_molecules, 2022, doi:10.3390/molecules27092942_

Round 1

Reviewer 1 Report

I have reviewed the manuscript entitled “Effects of Organic Acids on the Release of Fruity Esters in Water: An Insight at the Molecular Level”. The authors propose to investigate the role of OAs in ester solution and explore the mechanism insight into molecular level. It is well structured. There are a few deficiencies in this article need to be improved according to following list of comments:

General Comments:

OAs could change the pH of matrix which might affect aroma release, authors should add the relevant discussion.

Specific Comments:

1) 8: In Abstract, the first occurrence of an acronym requires a full name (OAs) .

2) 9: In Materials and Methods, what’s the basis of the added concentration of six esters?

3) 43-49: In Materials and Methods, The method of calculating the detection threshold needs to be labeled. BETs?

4) 51-59: In Materials and Methods, whether a conformational search was performed to determine the primary conformation pattern?

Reviewer 2 Report

please, see attached

Round 2

Reviewer 2 Report

The manuscript has improved. Here my last comment

Page 13, L44-45: ...probability of 0.5...? Please, complete the sentence. 

Author Response

Point 1: Page 13, L44-45: ...probability of 0.5...? Please, complete the sentence.

Response 1: We have completed the sentence.